

# Morphological connectivity correlates with trait impulsivity in healthy adults

Jingguang Li[1] and Xiang-Zhen Kong[2]

[1] College of Education, Dali University, Dali, China
[2] State Key Laboratory of Cognitive Neuroscience and Learning & IDG/McGovern Institute for Brain Research, Beijing Normal University, Beijing, China

## ABSTRACT

**Background**. Impulsivity is one crucial personality trait associated with various maladaptive behavior and many mental disorders. In the study reported here, we investigated the relationship between impulsivity and morphological connectivity (MC) between human brain regions, a newly proposed measure for brain coordination through the development and learning.

**Method**. Twenty-four participants' T1-weighted magnetic resonance imaging (MRI) images and their self-reported impulsivity scores, measured by the Barratt impulsiveness scale (BIS), were retrieved from the OpenfMRI project. First, we assessed the MC by quantifying the similarity of probability density function of local morphological features between the anterior cingulate cortex (ACC), one of the most crucial hubs in the neural network modulating cognitive control, and other association cortices in each participant. Then, we correlated the MC to impulsivity scores across participants.

**Results**. The BIS total score was found to correlate with the MCs between the ACC and two other brain regions in the right hemisphere: the inferior frontal gyrus (IFG), a well-established structure for inhibition control; the inferior temporal gyrus (ITG), which has been previously shown to be associated with hyperactive/impulsivity symptoms. Furthermore, the ACC-IFG MC was mainly correlated with motor impulsivity, and the ACC-ITG MC was mainly correlated with attentional impulsivity.

**Discussion**. Together, these findings provide evidence that the ACC, IFG, and ITG in the right hemisphere are involved neural networks modulating impulsivity. Also, the current findings highlight the utility of MC analyses in facilitating our understanding of neural correlates of behavioral and personality traits.

Corresponding author
Xiang-Zhen Kong,
kongxiangzheng@gmail.com,
xiangzhen.kong@outlook.com

## INTRODUCTION

Impulsivity, "a predisposition toward rapid, unplanned reactions to internal or external stimuli without regard to the negative consequences of these reactions to the impulsive individual or to others" (*Moeller et al., 2001*), is one crucial personality trait that may influence many facets of human life. Studies have suggested that high impulsivity is associated with risk for delinquency (*Lynam et al., 2000*), increased driving violations (*Owsley, McGwin & McNeal, 2003*), and more suicide attempts (*Dougherty et al., 2004*). Moreover, impulsivity has been shown to be a key characteristic of many psychiatric

disorders (*Moeller et al., 2001*), such as attention-deficit/hyperactivity disorder (*Winstanley, Eagle & Robbins, 2006*), drug dependence (*De Wit, 2009*), and borderline personality disorder (*Lieb et al., 2004*). In the study reported here, we investigated the neural subtracts underlying individual differences in impulsivity, which may provide invaluable insight into the mechanism of impulsive control and the development of treatments to prevent risky behaviors in individuals with high impulsivity.

To explore the neural correlates of trait impulsivity, researchers have succeeded linking the trait impulsivity to variability in brain structure and function using different neuroimaging techniques. First, the degree of damage within the right inferior frontal gyrus (IFG) has been associated with behavioral inhibition (*Aron et al., 2003*). Second, trait impulsivity has been associated with gray matter volume or cortical thickness in several brain regions, such as the orbitofrontal cortex (*Schilling et al., 2012*; *Matsuo et al., 2009*; *Crunelle et al., 2014*) and anterior cingulate cortex (ACC) (*Matsuo et al., 2009*; *Cho et al., 2013*; *Lee et al., 2013*). Third, trait impulsivity has been associated with variability of resting-state functional connectivity in neural networks, such as the amygdala network (*Xie et al., 2011*), default network (*Ding et al. 2013*), and whole-brain network architecture quantified by graph theory (*Davis et al., 2013*). Fourth, trait impulsivity has been associated with variability of white-matter integrity in several brain parts, such as anterior corpus callosum (*Moeller et al., 2005*), inferior frontal and anterior cingulate regions (*Romero et al., 2010*). In summary, these findings not only suggest that trait impulsivity reflects the function of different neural circuits, but also highlight the vital roles of certain structures (e.g., IFG, ACC, and OFC) in impulsive control.

The studies above have accumulated abundant evidence towards the neural correlates of impulsivity. However, no extant study has correlated impulsivity with morphological connectivity (MC), i.e., the covariance of regional gray matter morphology (e.g., volume or cortical thickness). The investigation of this brain-behavioral correlation would provide new insights into the neural basis of impulsivity. It is hypothesized that the morphological covariance between brain regions reflects synchronized development (*Alexander-Bloch, Giedd & Bullmore, 2013*), given that the development trajectory of the structural covariance is correlated the rate of change in cortical thickness and functional connectivity (*Raznahan et al., 2011*; *Alexander-Bloch et al., 2013*). Furthermore, the MC based on structural covariance in cortical thickness is associated with variability of mental functions. For example, higher IQ group shows larger MCs between the IFG and other frontal and parietal brain regions (*Lerch et al., 2006*). However, the covariance method is based on the correlation between subjects and thus can only be used with a large population, and it is incapable of measuring MCs for single subjects, which largely limits its application to individual differences studies in brain structure. To overcome the limitation, here we applied our newly-proposed approach to measuring MC for each individual from the MRI data (*Kong et al., 2014*). The new approach is based on similarity of regional morphological distributions and has shown its excellent test-retest reliability and validation reflecting human brain architecture and neuroplasticity (*Kong et al., 2014*; *Kong et al., 2015*; *Wang et al., 2016*).

Given that no previous studies have tested the neural correlates of impulsivity using the MC approach, here we conducted an exploratory investigation by correlating trait impulsivity to individual differences in MC in a public MRI data. First, we measured the MCs of interest for each individual from the MRI data. Specifically, we focused on the MCs of the ACC. The ACC was selected as the seed point for three reasons: (i) the ACC is one of most widely known region for cognitive control, especially in conflict monitoring (*Bush, Luu & Posner, 2000*; *Botvinick, Cohen & Carter, 2004*); (ii) the ACC plays a central role in receiving convergent inputs from multiple cortical regions and is considered to be involved in more complex and integrated cognitive activities including executive functions (*Carter et al., 2000*; *Koo et al., 2008*; *Mesulam, 2000*; *Rosen et al., 2005*); (iii) moreover, the gray matter of the ACC showed the most apparent association with individual differences in impulsivity (*Cho et al., 2013*; *Lee et al., 2013*). Then, correlation analysis was used to investigate the possible associations between the MCs of interest and the scores of impulsivity.

## MATERIALS AND METHODS

### Dataset

#### Participants

Twenty-four right-handed participants (mean age: 20.8 years, range: 18–33 years; 10 females) were included in this study. This data was obtained from the OpenfMRI database. Its accession number is ds000009. Participants reported no past or current psychiatric illness or history of neurological disorders. All participants provided written informed consent according to the procedures of the UCLA Institutional Review Board. In this study, we mainly used the T1-weighted MRI images and the behavioral assessment of impulsivity (see below). For more detail about the dataset, please see *Cohen & Poldrack (2014)*.

#### Assessment of impulsivity

Impulsivity was assessed using the Barratt Impulsiveness Scale, Version 11 (BIS-11; *Patton & Stanford, 1995*), which was a 4-point Likert-type scale containing 30 items. In the BIS, higher values indicate greater levels of impulsivity. In addition to the total score as a comprehensive measure of impulsivity, BIS contains three components of impulsivity: attentional (rapid shifts and impatience with complexity), motor (impetuous action) and non-planning (lack of future orientation) impulsivity. Similarly, higher scores demonstrate higher impulsivity.

#### MRI scanning

Imaging data were collected with a 3T Siemens Trio scanner at the Ahmanson-Lovelace Brain Mapping Center at the University of California, Los Angeles. For each participant, a magnetization prepared rapid acquisition gradient echo (MP-RAGE) MRI sequence (176 sagittal slices, slice thickness 1 mm, TR = 1,900 ms, TE = 2.26 ms, matrix 256 × 256, field of view 250) was used to obtain the T1-weighted image of the entire brain.

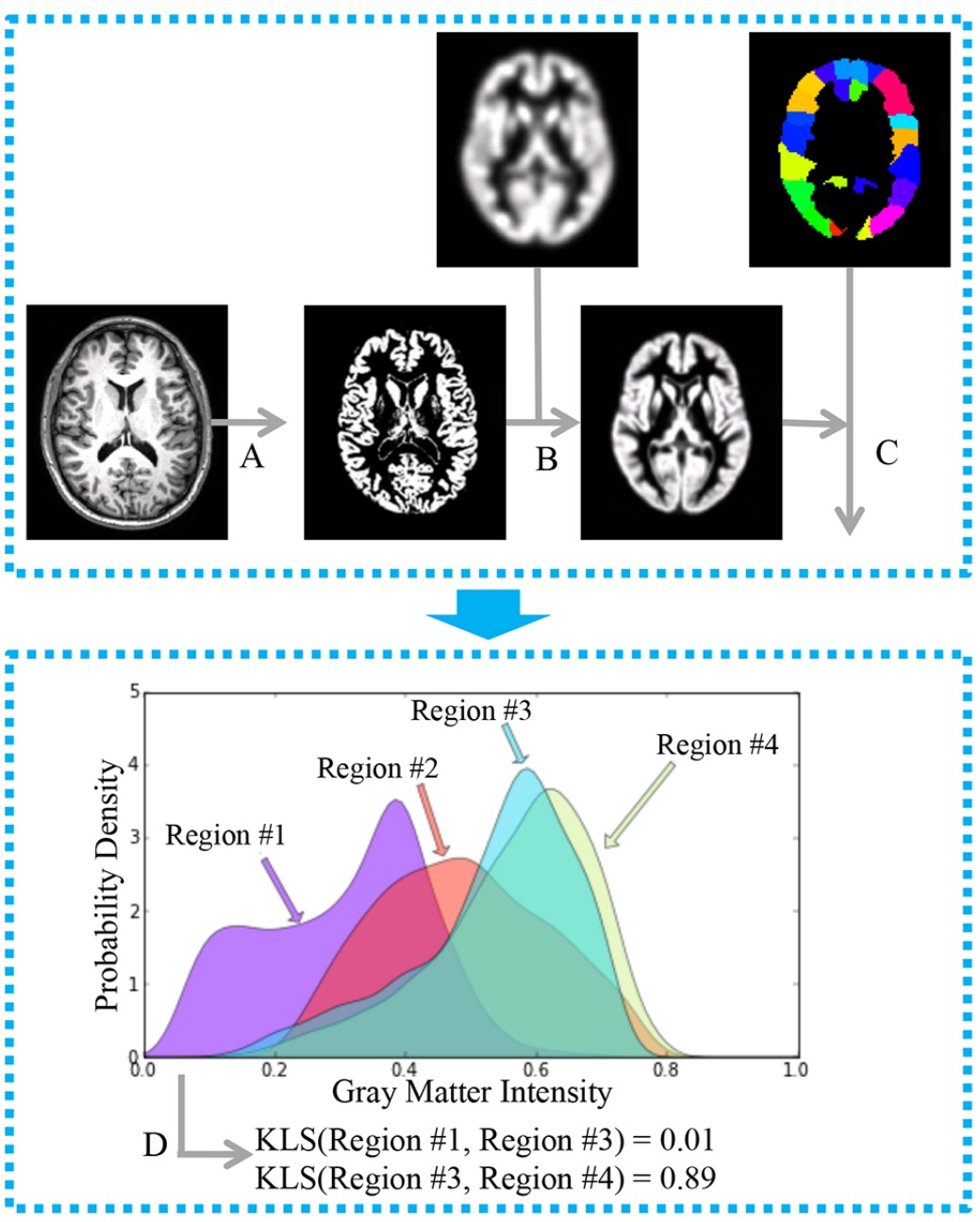

**Figure 1** **General workflow for the estimation of inter-regional morphological connectivity from single-subject MRI data.** (A) MR images were segmented to create gray matter (GM) images; (B) GM images for each individual were normalized to the standard template in MNI152 space, and then modulated and smoothed for further analysis. (C) Brain parcellation and estimation of regional probability density functions (PDFs). Bottom panel: (D) Morphological connectivity (MC) between PDFs from different brain regions was quantified.

## Morphological connectivity

As shown in Fig. 1, the calculation of morphological connectivity (*Kong et al., 2014*) can be summarized in the following three steps:

  (1) Calculation of local morphological features for each voxel in the brain.
  (2) Definition of ROIs and estimation of the morphological distribution for each ROI.
  (3) Quantification of the similarity of the morphological distributions of two ROIs as a measure of the morphological connectivity.

Specifically, step 1 is done with an automatic neuroimaging technique called voxel-based morphometry (VBM) (*Ashburner & Friston, 2000*) implemented in Statistical Parametric Mapping version 8 (SPM8, http://www.fil.ion.ucl.ac.uk/spm/). The default parameters were applied for estimating regional morphological feature, including the Diffeomorphic Anatomical Registration Through Exponential Lie Algebra (DARTEL) approach (*Ashburner, 2007*) and an 8-mm full-width half-maximum (FWHM). In step 2, for each ROI, the regional probability density function (PDF) was estimated using kernel density estimation (KDE) implemented in the Scipy package (gaussian_kde with the Scott's Rule; http://www.scipy.org/). A fixed gray matter intensity boundary [0, 1] was used for the estimation. Finally, in step 3, the MC for each pair of ROIs was calculated using a similarity metric based on the Kullback–Leibler (KL) divergence. The MC ranges from 0 to 1, where 1 is for two identical distributions. For more technical detail, please see *Kong et al. (2014)*.

## ROI selection

In this study, we focused on the morphological relations of the anterior cingulate cortex (ACC) with the cortical association cortices. Specifically, we identified bilateral ACC and cortical association ROIs (22 for each hemisphere) with the widely used AAL atlas (*Tzourio-Mazoyer et al., 2002*). For each hemisphere, these ROIs were included: Superior frontal gyrus, dorsolateral (SFGdor), Middle frontal gyrus (MFG), Inferior frontal gyrus, opercular part (IFGoper), Inferior frontal gyrus, triangular part (IFGtri), Rolandic operculum (ROL), Supplementary motor area (SMA), Superior frontal gyrus, medial (SFGmed), Cuneus (CUN), Lingual gyrus (LING), Superior occipital gyrus (SOG), Middle occipital gyrus (MOG), Inferior occipital gyrus (IOG), Fusiform gyrus (FFG), Superior parietal gyrus (SPG), Inferior parietal, but supramarginal and angular gyri (IPL), Supramarginal gyrus (SMG), Angular gyrus (ANG), Precuneus (PCUN), Paracentral lobule (PCL), Superior temporal gyrus (STG), Middle temporal gyrus (MTG), Inferior temporal gyrus (ITG). The association cortices include most of the cerebral surface of the human brain and are largely responsible for the complex processing, including impulsive behavior control. In addition, focusing on these regions would reduce the multiple comparisons problem.

## Statistical analysis

The MCs were calculated for each hemisphere for each participant. To investigate the behavioral relevance of the MC, we related the individual differences in MC to the variability in impulsivity with partial correlation analysis, controlling for age and sex. Since multiple comparisons were performed in the analysis, a significant threshold of $p < 0.05$ (FDR

**Table 1  Results of impulsivity scores measured by the Barratt impulsiveness scale (BIS).**

|  | Mean (SD) | Range |
|---|---|---|
| Total score of BIS | 63.7 (8.4) | 47–91 |
| Non-planning impulsivity | 24.1 (3.8) | 16–35 |
| Attentional impulsivity | 17.5 (3.2) | 11–26 |
| Motor impulsivity | 22.1 (4.1) | 14–30 |

corrected) was applied. Given the multi-dimensional property of impulsivity (*Stanford et al., 2009*), we further investigated which components of impulsivity (attentional, motor, and non-planning impulsivity) mainly contributed to the observed associations. The standard error (SE) and 95% confidence interval (CI) were calculated for each correlation based on Fisher's $Z$ transformation using the CIr function from R package *psychometric*. Besides the partial correlation analysis, mediation analysis (*MacKinnon, Fairchild & Fritz, 2007*) was used to determine whether some facet of impulsivity could fully explain to the observed associations.

### Control analysis for head size
Head size is another possible confounding source in MRI studies, especially in the analysis of cortical volume (*Barnes et al., 2010*). Thus, to rule out the possibility that the observed associations between inter-regional MC and impulsivity measures were accounted for by this general factor, we estimated head size based on the MRI data using FreeSurfer (https://surfer.nmr.mgh.harvard.edu/). After obtaining head size for each individual, we examined whether the observed associations remained after controlling for the head size. In this control analysis, age and sex were controlled as did in the main analyses.

## RESULTS
As expected, both the total impulsivity score and its three components showed wide variability in performance noted across participants (Table 1).

To investigate how the variability of impulsivity relates to the MC of the ACC, we first quantified the MC value between the ACC and each cortical region with the newly-proposed method (*Kong et al., 2014*) for each participant. Then, we conducted association analyses separately for each hemisphere (that is, only ipsilateral relations were considered) to investigate the possible associations. Given that only a few structural connections (∼2%) exist between two hemispheres, most of which mainly connects homologous regions (*Abeles, 1991*), here we focused on ipsilateral connections with the ACC. We observed specific associations of total impulsivity score with two morphological relations that survived multiple comparisons correction ($p < 0.05$, FDR corrected) in the right hemisphere (Fig. 2A): the morphological connectivity between the ACC and the IFGtri (hereafter referred to as ACC-IFG MC) and the morphological connectivity between the ACC and the ITG (hereafter referred to as ACC-ITG MC). Specifically, individual differences in impulsivity showed a positive association with the ACC-IFG MC (Fig. 2B; $r = 0.64, p = 0.001$, SE $= 0.097$; 95% CI [0.32, 0.83]; Spearman's rho $= 0.43, p = 0.035$), and

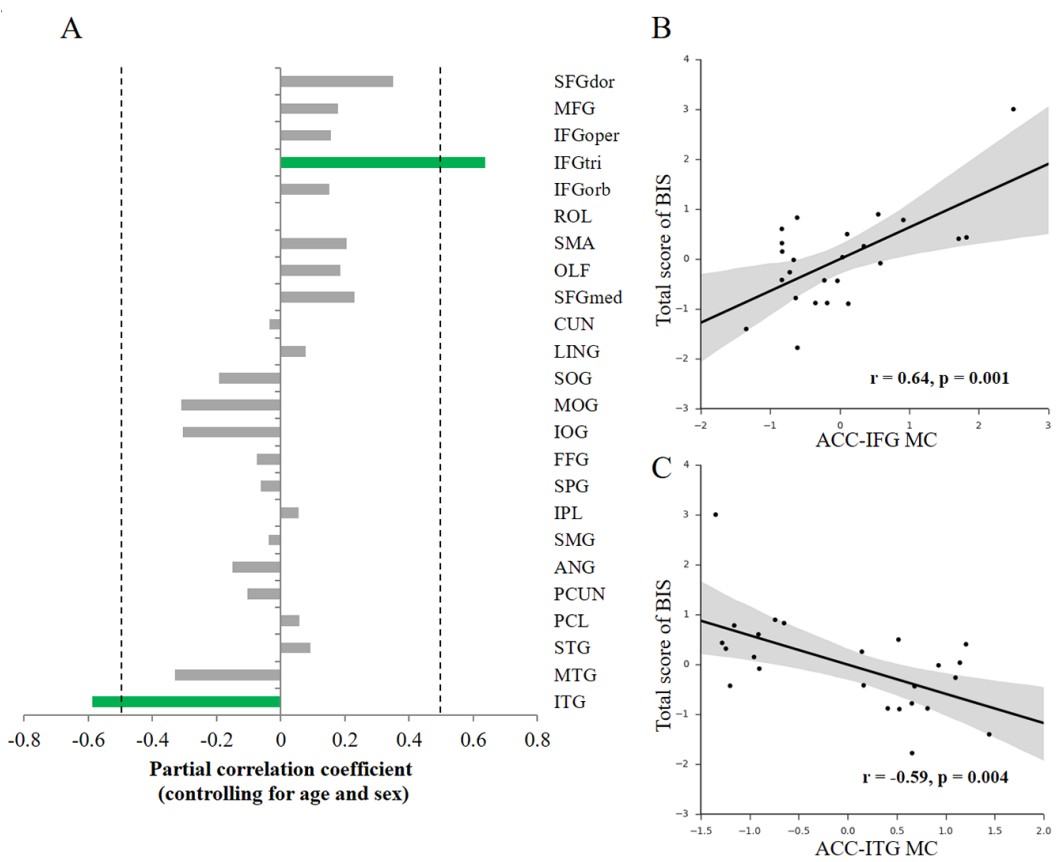

**Figure 2** **Behavior-morphological connectivity (MC) correlations.** (A) The *x*-axis denotes the correlation coefficient for the behavior–MC correlation (age and sex controlled). The dashed line indicates the significance level of the behavior–MC correlation analysis ($p < 0.05$, uncorrected). The green bar indicates correlations that survived the multiple comparisons correction ($p < 0.05$, FDR-corrected). (B) Scatter plot represents the linear association between impulsivity and the MC values of the ACC and IFGtri (i.e., ACC-IFG MC). (C) Scatter plot represents the linear association between impulsivity and the MC values of the ACC and ITG (i.e., ACC-ITG MC). Each axis shows the residual score after controlling for age and sex; shaded regions depict 95% confidence intervals.

showed a negative association with the ACC-ITG MC (Fig. 2C; $r = -0.59$, $p = 0.004$, SE = 0.18; 95% CI $[-0.80, -0.24]$; Spearman's rho = $-0.53$, $p = 0.008$). In addition, to test the potential influences of variations in confounding variables included in the above correlation analyses (i.e., age or sex), we ran additional analyses with additional confounding variables settings (*Van Schuerbeek, Baeken & De Mey, 2016*). The results remain significant with either sex (ACC-IFG MC: $r = 0.64$, $p = 0.001$; ACC-ITG MC: $r = -0.53$, $p = 0.009$), age (ACC-IFG MC: $r = 0.61$, $p = 0.002$; ACC-ITG MC: $r = -0.47$, $p = 0.022$), or none of them (ACC-IFG MC: $r = 0.60$, $p = 0.002$; ACC-ITG MC: $r = -0.47$, $p = 0.022$) being included in the correlation analysis.

Moreover, no significant association was found in the left hemisphere, though a similar trend was observed with the MC between the left ACC and left IFG ($r = 0.41$, $p = 0.059$, uncorrected). Thus, the ACC-IFG MC and ACC-ITG MC were focused in the following analyses.

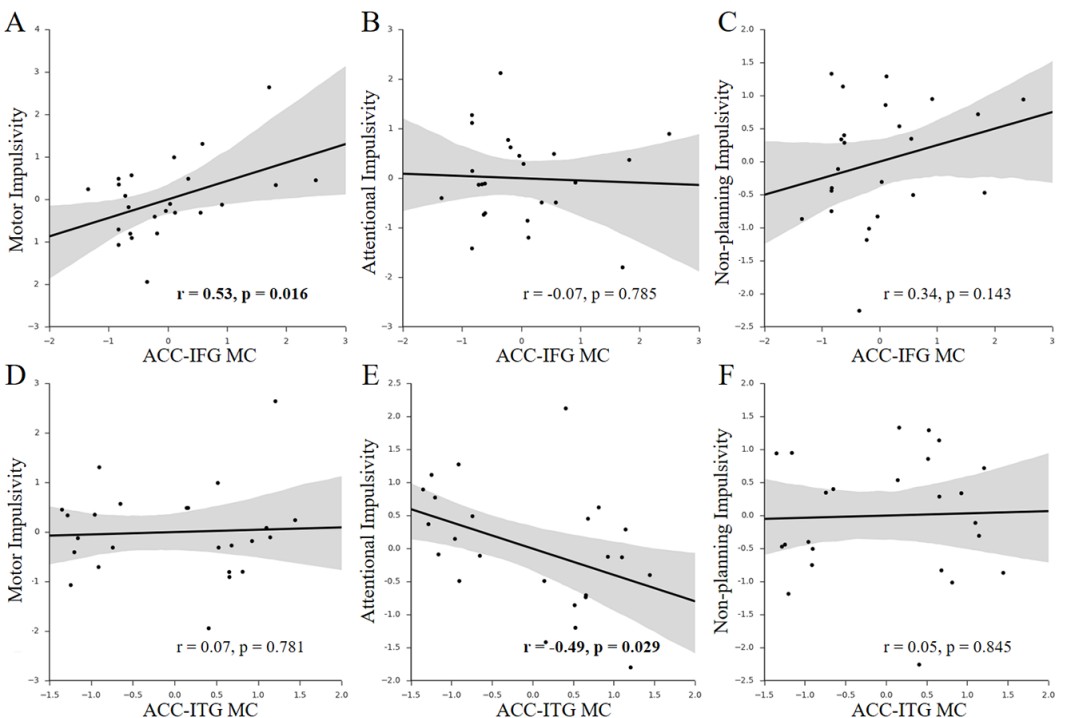

**Figure 3** **Behavior-MC correlations with facets of impulsivity.** (A–C) Scatter plots for the associations between facets of impulsivity and the ACC-IFG MC. (D–F) Scatter plots for the associations between facets of impulsivity and the ACC-ITG MC. Figures **in bold** indicate that the statistic is significant ($p < 0.05$); each axis shows the residual score after controlling for age and sex; shaded regions depict 95% confidence intervals.

Given the multi-dimensional property of impulsivity (*Stanford et al., 2009*), we further investigated which components of impulsivity mainly contributed to the observed associations. We found that only the motor impulsivity was significantly associated with the ACC-IFG MC (Figs. 3A–3C; motor impulsivity: $r = 0.53$, $p = 0.016$, SE = 0.12, 95% CI [0.16, 0.77]; attentional impulsivity: $r = -0.07$, $p = 0.785$, SE = 0.21, 95% CI [−0.46, 0.34]; non-planning impulsivity: $r = 0.34$, $p = 0.143$, SE = 0.16, 95% CI [−0.073, 0.65]). As expected, mediation analysis showed that the motor impulsivity fully mediated the association between impulsivity and the ACC-IFG MC (beta decreased from 0.64 to 0.33; $p < 0.05$; 95% confidence interval (CI): [0.08, 0.62]), suggesting that the observed association with the ACC-IFG MC was mainly contributed by the motor facet of impulsivity. Furthermore, we found that only the attentional impulsivity was significantly associated with the ACC-ITG MC (Figs. 3D–3F; motor impulsivity: $r = 0.07$, $p = 0.781$, SE = 0.20, 95% CI [−0.34, 0.46]; attentional impulsivity: $r = -0.49$, $p = 0.029$, SE = 0.19, 95% CI [−0.75, −0.11]; non-planning: $r = 0.05$, $p = 0.845$, SE = 0.20, 95% CI [−0.36, 0.44]). Similarly, mediation analysis showed that the attentional impulsivity fully mediated the association between impulsivity and the ACC-ITG MC (beta decreased from 0.66 to 0.04; $p < 0.05$; 95% CI [−1.22, 0.22]), suggesting that the observed association with the ACC-ITG MC was mainly contributed by the attentional facet of impulsivity. Taken together, the

attentional impulsivity and motor impulsivity were mainly associated with ACC-IFG MC and ACC-ITG MC, respectively.

Finally, control analysis was performed to rule out potential confounding effects from head size (see 'Materials and Methods'). We found that the observed associations between MCs and impulsivity measures remained after controlling for head size (total impulsivity scores & ACC-IFG MC: $r = 0.67$, $p = 0.001$; total impulsivity scores & ACC-ITG MC: $r = -0.57$, $p = 0.007$; motor impulsivity & ACC-IFG MC: $r = 0.60$, $p = 0.004$; attentional impulsivity & ACC-ITG MC: $-0.65$, $p = 0.002$), suggesting that our findings were unlikely contributed by head size.

## DISCUSSION

This is the first study on structural correlates of trait impulsivity focusing on MC. We found that the ACC-IFG MC and ACC-ITG MC in the right hemisphere were correlated to scores of self-reported impulsivity. Furthermore, two dimensions of the impulsivity, i.e., BIS attentional and BIS motor impulsivity, were correlated with to the ACC-IFG MC and ACC-ITG MC, respectively.

Our findings on the right IFG are consistent with the extant literature. It is well-established that the right IFG is critical for inhibition control (see *Aron, Robbins, & Poldrack (2004)* and *Aron, Robbins, & Poldrack (2014)* for reviews). A meta-analysis of eighteen fMRI studies suggests that Go/No-go task elicits the largest activation in the right IFG among other brain regions (*Buchsbaum et al., 2005*). Also, lesion of the right IFG is associated with deficits in performance in stop-signal and Go/No-go tasks (*Aron et al., 2003*), and temporary deactivation of the right IFG by transcranial magnetic stimulation would selectively impair the stop-signal response (*Chambers et al. 2006*). Therefore, our finding that the ACC-IFG MC was correlated to BIS total score provides new converging evidence to support the role of right IFG in inhibition control.

Interestingly, the BIS motor impulsivity (sample item: "I buy things on impulse"), defined as "acting without thinking", nicely fitted with the concept with inhibition control. However, the BIS attentional (sample item: "I am restless at the theater or lectures") and non-planning impulsivity (sample item: "I plan tasks carefully") appear to less related to inhibition control. In accordance with this speculation, the correlation between the BIS motor impulsivity and behavioral inhibition performance (Go/No-go paradigm) is larger than the correlations between the other two BIS subscale scores (i.e., attentional and non-planning impulsivity) and behavioral inhibition performance (*Keilp, Sackeim & Mann, 2005*; *Spinella, 2004*). The separate functions of BIS subscales, in conjunction with the functionality of right IFG, fitted well with the finding in our study: BIS motor impulsivity, but not other BIS subscale scores, was mainly correlated with ACC-IFG MC.

Our findings on the right ITG is somewhat unexpected because it is not a classical region for impulsive control in the neuroscience literature. However, one study has observed that the rate of thinning of cortical thickness in the right ITG was associated with hyperactive/impulsive symptoms in developing children (*Shaw et al., 2011*). Consistent with this finding, we found that the ACC-ITG MC was associated with BIS total and
attentional impulsivity scores. Taken together, these findings implied that the right ITG might also be a part of the neural network for impulsive control.

Our work could be extended in several ways. First, other local measures of gray matter, including cortical thickness and cortical surface area, could be used in a similar way to the current study. Second, another investigation by us has demonstrated that it is plausible to use graph theory to explore the inter-individual differences in MC at a whole-brain network level, and that both the IFG and ITG have been identified as hubs (*Kong et al., 2015*). Thus, it is intriguing to correlate the trait impulsivity to graph-based properties, which may advance our understanding of the neural circuits of impulsive control in a more holistic manner. Third, high impulsivity is a crucial characteristic of many psychiatric disorders (*Moeller et al., 2001*). Therefore, future studies of MC in psychiatric patients with problems of impulsive control would help to elucidate the pathophysiology of these disorders.

Finally, several methodological limitations of the current study deserve consideration. First, because the BIS is a self-report scale, it might only partly capture impulsivity. Future studies could include cognitive tasks such as Go/No go, Stop-signal tasks, and relate the task performance to trait impulsivity. Second, because the sample size is relatively small, the correlation findings would be sensitive to outliers and need to be viewed with some caution. Future studies with large sample size are needed to replicate the findings and may detect more subtle relationships with more statistical power. Third, a recent study (*Van Schuerbeek, Baeken & De Mey, 2016*) has suggested that linking personality traits to brain structure would be sensitive to methodological choices, including the regression model used and ROI labeling. To test the robustness of our findings against variations in confounding variables (i.e., age or sex) included in the correlation analysis, we ran additional analyses with different confounding variables settings, but the results remain unchanged. However, when another atlas was used for the ROI definition (i.e., the Desikan-Killiany atlas from FreeSurfer), we failed to replicate the correlation with ACC-IFG MC, although the correlation with ACC-ITG MC remained significant. The lack of replication of the correlation about with the IFG could largely be due to the difference of the brain parcellation (e.g., 1,575 voxels in AAL atlas versus 609 voxels in Desikan-Killiany atlas), and suggested the potential parcellation-dependent effects in connectivity-related methods (*Wang et al., 2009*; *Arslan et al., in press*). Given there is no optimal method able to address this challenge so far, replication studies with independent samples will be important to confirm these findings.

## CONCLUSIONS

This is the first study that investigated brain morphological connectivity patterns underlying impulsivity in healthy individuals. Our results indicate that the ACC-IFG MC and ACC-ITG MC are associated with trait impulsivity in healthy individuals and further supported that the ACC, IFG, and ITG in the right hemisphere were involved a network modulating impulsive control. Moreover, the present data provided a demonstration that the MC varies as a function of trait impulsivity, which highlighted the utility of MC analyses in facilitating our understanding neural correlates of behavioral and personality traits.

### Funding

This work was supported by the National Natural Science Foundation of China (No. 31500884) and the Key Projects of Yunnan Provincial Department of Education (No. 2015Z148). The funders had no role in study design, data collection and analysis, decision to publish, or preparation of the manuscript.

### Grant Disclosures

The following grant information was disclosed by the authors:
National Natural Science Foundation of China: 31500884.
Key Projects of Yunnan Provincial Department of Education: 2015Z148.

### Competing Interests

The authors declare there are no competing interests.

### Author Contributions

- Jingguang Li analyzed the data, contributed reagents/materials/analysis tools, wrote the paper, prepared figures and/or tables, reviewed drafts of the paper.
- Xiang-Zhen Kong conceived and designed the experiments, analyzed the data, contributed reagents/materials/analysis tools, wrote the paper, prepared figures and/or tables, reviewed drafts of the paper.

### Human Ethics

The following information was supplied relating to ethical approvals (i.e., approving body and any reference numbers):

This data was obtained from the OpenfMRI database (accession number ds000009). All participants provided written informed consent according to the procedures of the UCLA Institutional Review Board.

No approval number is available from the description of this open data (Cohen and Poldrack, 2014: Cohen JR, and Poldrack RA. 2014. Materials and methods for OpenfMRI ds009: the generality of self control. Available at https://openfmri.org/media/ds000009/ds009_methods_0_CchSZHn.pdf).

### Data Availability

This data was obtained from the OpenfMRI database (accession number ds000009). For details, see Cohen JR, and Poldrack RA. 2014: Materials and methods for OpenfMRI ds009: the generality of self control. Available at https://openfmri.org/media/ds000009/ds009_methods_0_CchSZHn.pdf.

## REFERENCES

**Abeles M. 1991.** *Corticonics: neural circuits of the cerebral cortex.* Cambridge: Cambridge University Press.

# PeerJ

**Alexander-Bloch A, Giedd JN, Bullmore E. 2013.** Imaging structural co-variance between human brain regions. *Nature Reviews Neuroscience* **14**:322–336 DOI 10.1038/nrn3465.

**Alexander-Bloch A, Raznahan A, Bullmore E, Giedd J. 2013.** The convergence of maturational change and structural covariance in human cortical networks. *Journal of Neuroscience* **33**:2889–2899 DOI 10.1523/JNEUROSCI.3554-12.2013.

**Aron AR, Fletcher PC, Bullmore ET, Sahakian BJ, Robbins TW. 2003.** Stop-signal inhibition disrupted by damage to right inferior frontal gyrus in humans. *Nature Neuroscience* **6**:115–116 DOI 10.1038/nn1003.

**Aron AR, Robbins TW, Poldrack RA. 2004.** Inhibition and the right inferior frontal cortex. *Trends in Cognitive Sciences* **8**:170–177 DOI 10.1016/j.tics.2004.02.010.

**Aron AR, Robbins TW, Poldrack RA. 2014.** Inhibition and the right inferior frontal cortex: one decade on. *Trends in Cognitive Sciences* **18**:177–185 DOI 10.1016/j.tics.2013.12.003.

**Arslan S, Ktena SI, Makropoulos A, Robinson EC, Rueckert D, Parisot S. 2017.** Human brain mapping: a systematic comparison of parcellation methods for the human cerebral cortex. *NeuroImage* In Press DOI 10.1016/j.neuroimage.2017.04.014.

**Ashburner J. 2007.** A fast diffeomorphic image registration algorithm. *Neuroimage* **38(1)**:95–113.

**Ashburner J, Friston KJ. 2000.** Voxel-based morphometry—the methods. *Neuroimage* **11**:805–821 DOI 10.1006/nimg.2000.0582.

**Barnes J, Ridgway GR, Bartlett J, Henley SM, Lehmann M, Hobbs N, Clarkson MJ, MacManus DG, Ourselin S, Fox NC. 2010.** Head size, age and gender adjustment in MRI studies: a necessary nuisance? *Neuroimage* **53**:1244–1255.

**Botvinick MM, Cohen JD, Carter CS. 2004.** Conflict monitoring and anterior cingulate cortex: an update. *Trends in Cognitive Sciences* **8**:539–546 DOI 10.1016/j.tics.2004.10.003.

**Buchsbaum BR, Greer S, Chang WL, Berman KF. 2005.** Meta-analysis of neuroimaging studies of the Wisconsin Card-Sorting task and component processes. *Human Brain Mapping* **25**:35–45 DOI 10.1002/hbm.20128.

**Bush G, Luu P, Posner MI. 2000.** Cognitive and emotional influences in anterior cingulate cortex. *Trends in Cognitive Sciences* **4**:215–222 DOI 10.1016/S1364-6613(00)01483-2.

**Carter CS, Macdonald AM, Botvinick M, Ross LL, Stenger VA, Noll D, Cohen JD. 2000.** Parsing executive processes: strategic vs. evaluative functions of the anterior cingulate cortex. *Proceedings of the National Academy of Sciences of the United States of America* **97**:1944–1948 DOI 10.1073/pnas.97.4.1944.

**Chambers CD, Bellgrove MA, Stokes MG, Henderson TR, Garavan H, Robertson IH, Morris AP, Mattingley JB. 2006.** Executive "brake failure" following deactivation of human frontal lobe. *Journal of Cognitive Neuroscience* **18**:444–455.

**Cho SS, Pellecchia G, Aminian K, Ray N, Segura B, Obeso I, Strafella AP. 2013.** Morphometric correlation of impulsivity in medial prefrontal cortex. *Brain Topography* **26**:479–487 DOI 10.1007/s10548-012-0270-x.

**Cohen JR, Poldrack RA. 2014.** Materials and methods for OpenfMRI ds009: the generality of self control. *Available at https://openfmri.org/media/ds000009/ds009_methods_0_CchSZHn.pdf* (accessed on 01 Feb 2017).

**Crunelle CL, Kaag AM, Van Wingen G, Van den Munkhof HE, Homberg JR, Reneman L, Van den Brink W. 2014.** Reduced frontal brain volume in non-treatment-seeking cocaine-dependent individuals: exploring the role of impulsivity, depression, and smoking. *Frontiers in Human Neuroscience* **8**:Article 7 DOI 10.3389/fnhum.2014.00007.

**Davis FC, Knodt AR, Sporns O, Lahey BB, Zald DH, Brigidi BD, Hariri AR. 2013.** Impulsivity and the modular organization of resting-state neural networks. *Cerebral Cortex* **23**:1444–1452 DOI 10.1093/cercor/bhs126.

**De Wit H. 2009.** Impulsivity as a determinant and consequence of drug use: a review of underlying processes. *Addiction Biology* **14**:22–31 DOI 10.1111/j.1369-1600.2008.00129.x.

**Ding W-n, Sun J-h, Sun Y-w, Zhou Y, Li L, Xu J-r, Du Y-s. 2013.** Altered default network resting-state functional connectivity in adolescents with Internet gaming addiction. *PLOS ONE* **8**:e59902 DOI 10.1371/journal.pone.0059902.

**Dougherty DM, Mathias CW, Marsh DM, Papageorgiou TD, Swann AC, Moeller FG. 2004.** Laboratory measured behavioral impulsivity relates to suicide attempt history. *Suicide and Life-Threatening Behavior* **34**:374–385 DOI 10.1521/suli.34.4.374.53738.

**Keilp JG, Sackeim HA, Mann JJ. 2005.** Correlates of trait impulsiveness in performance measures and neuropsychological tests. *Psychiatry Research* **135**:191–201 DOI 10.1016/j.psychres.2005.03.006.

**Kong X-Z, Liu Z, Huang L, Wang X, Yang Z, Zhou G, Zhen Z, Liu J. 2015.** Mapping individual brain networks using statistical similarity in regional morphology from MRI. *PLOS ONE* **10**:e0141840 DOI 10.1371/journal.pone.0141840.

**Kong X-Z, Wang X, Huang L, Pu Y, Yang Z, Dang X, Zhen Z, Liu J. 2014.** Measuring individual morphological relationship of cortical regions. *Journal of Neuroscience Methods* **237**:103–107 DOI 10.1016/j.jneumeth.2014.09.003.

**Koo M-S, Levitt JJ, Salisbury DF, Nakamura M, Shenton ME, McCarley RW. 2008.** A cross-sectional and longitudinal magnetic resonance imaging study of cingulate gyrus gray matter volume abnormalities in first-episode schizophrenia and first-episode affective psychosis. *Archives of General Psychiatry* **65**:746–760 DOI 10.1001/archpsyc.65.7.746.

**Lee TY, Kim SN, Jang JH, Shim G, Jung WH, Shin NY, Kwon JS. 2013.** Neural correlate of impulsivity in subjects at ultra-high risk for psychosis. *Progress in Neuro-Psychopharmacology and Biological Psychiatry* **45**:165–169 DOI 10.1016/j.pnpbp.2013.04.008.

**Lerch JP, Worsley K, Shaw WP, Greenstein DK, Lenroot RK, Giedd J, Evans AC. 2006.** Mapping anatomical correlations across cerebral cortex (MACACC) using cortical thickness from MRI. *Neuroimage* **31**:993–1003 DOI 10.1016/j.neuroimage.2006.01.042.

**Lieb K, Zanarini MC, Schmahl C, Linehan MM, Bohus M. 2004.** Borderline personality disorder. *The Lancet* **364**:453–461 DOI 10.1016/S0140-6736(04)16770-6.

**Lynam DR, Caspi A, Moffit TE, Wikström P-O, Loeber R, Novak S. 2000.** The interaction between impulsivity and neighborhood context on offending: the effects of impulsivity are stronger in poorer neighborhoods. *Journal of Abnormal Psychology* **109**:563–574 DOI 10.1037/0021-843X.109.4.563.

**MacKinnon DP, Fairchild AJ, Fritz MS. 2007.** Mediation analysis. *Annual Review of Psychology* **58**:593–614 DOI 10.1146/annurev.psych.58.110405.085542.

**Matsuo K, Nicoletti M, Nemoto K, Hatch JP, Peluso MA, Nery FG, Soares JC. 2009.** A voxel-based morphometry study of frontal gray matter correlates of impulsivity. *Human Brain Mapping* **30**:1188–1195 DOI 10.1002/hbm.20588.

**Mesulam M-M. 2000.** *Principles of behavioral and cognitive neurology.* Oxford: Oxford University Press.

**Moeller FG, Barratt ES, Dougherty DM, Schmitz JM, Swann AC. 2001.** Psychiatric aspects of impulsivity. *American Journal of Psychiatry* **158**:1783–1793 DOI 10.1176/appi.ajp.158.11.1783.

**Moeller FG, Hasan KM, Steinberg JL, Kramer LA, Dougherty DM, Santos RM, Valdes I, Swann AC, Barratt ES, Narayana PA. 2005.** Reduced anterior corpus callosum white matter integrity is related to increased impulsivity and reduced discriminability in cocaine-dependent subjects: diffusion tensor imaging. *Neuropsychopharmacology* **30**:610–617 DOI 10.1038/sj.npp.1300617.

**Owsley C, McGwin G, McNeal SF. 2003.** Impact of impulsiveness, venturesomeness, and empathy on driving by older adults. *Journal of Safety Research* **34**:353–359 DOI 10.1016/j.jsr.2003.09.013.

**Patton JH, Stanford MS. 1995.** Factor structure of the Barratt impulsiveness scale. *Journal of Clinical Psychology* **51**:768–774 DOI 10.1002/1097-4679(199511)51:6<768::AID-JCLP2270510607>3.0.CO;2-1.

**Raznahan A, Lerch JP, Lee N, Greenstein D, Wallace GL, Stockman M, Clasen L, Shaw PW, Giedd JN. 2011.** Patterns of coordinated anatomical change in human cortical development: a longitudinal neuroimaging study of maturational coupling. *Neuron* **72**:873–884 DOI 10.1016/j.neuron.2011.09.028.

**Romero MJ, Asensio S, Palau C, Sanchez A, Romero FJ. 2010.** Cocaine addiction: diffusion tensor imaging study of the inferior frontal and anterior cingulate white matter. *Psychiatry Research: Neuroimaging* **181**:57–63 DOI 10.1016/j.pscychresns.2009.07.004.

**Rosen HJ, Allison SC, Schauer GF, Gorno-Tempini ML, Weiner MW, Miller BL. 2005.** Neuroanatomical correlates of behavioural disorders in dementia. *Brain* **128**:2612–2625 DOI 10.1093/brain/awh628.

**Schilling C, Kühn S, Romanowski A, Schubert F, Kathmann N, Gallinat J. 2012.** Cortical thickness correlates with impulsiveness in healthy adults. *Neuroimage* **59**:824–830 DOI 10.1016/j.neuroimage.2011.07.058.

**Shaw P, Gilliam M, Liverpool M, Weddle C, Malek M, Sharp W, Greenstein D, Evans A, Rapoport J, Giedd J. 2011.** Cortical development in typically developing children with symptoms of hyperactivity and impulsivity: support for a dimensional view of attention deficit hyperactivity disorder. *American Journal of Psychiatry* **168**:143–151 DOI 10.1176/appi.ajp.2010.10030385.

**Spinella M. 2004.** Neurobehavioral correlates of impulsivity: evidence of prefrontal involvement. *International Journal of Neuroscience* **114**:95–104 DOI 10.1080/00207450490249347.

**Stanford MS, Mathias CW, Dougherty DM, Lake SL, Anderson NE, Patton JH. 2009.** Fifty years of the Barratt Impulsiveness Scale: An update and review. *Personality and Individual Differences* **47**:385–395 DOI 10.1016/j.paid.2009.04.008.

**Tzourio-Mazoyer N, Landeau B, Papathanassiou D, Crivello F, Etard O, Delcroix N, Mazoyer B, Joliot M. 2002.** Automated anatomical labeling of activations in SPM using a macroscopic anatomical parcellation of the MNI MRI single-subject brain. *Neuroimage* **15**:273–289 DOI 10.1006/nimg.2001.0978.

**Van Schuerbeek P, Baeken C, De Mey J. 2016.** The heterogeneity in retrieved relations between the personality trait 'harm avoidance' and gray matter volumes due to variations in the VBM and ROI labeling processing settings. *PLOS ONE* **11(4)**:e0153865 DOI 10.1371/journal.pone.0153865.

**Wang H, Jin X, Zhang Y, Wang J. 2016.** Single-subject morphological brain networks: connectivity mapping, topological characterization and test–retest reliability. *Brain and Behavior* **6(4)**:e00448 DOI 10.1002/brb3.448.

**Wang J, Wang L, Zang Y, Yang H, Tang H, Gong Q, Chen Z, Zhu C, He Y. 2009.** Parcellation-dependent small-world brain functional networks: a resting-state fMRI study. *Human Brain Mapping* **30(5)**:1511–1523 DOI 10.1002/hbm.20623.

**Winstanley CA, Eagle DM, Robbins TW. 2006.** Behavioral models of impulsivity in relation to ADHD: translation between clinical and preclinical studies. *Clinical Psychology Review* **26**:379–395 DOI 10.1016/j.cpr.2006.01.001.

**Xie C, Li S-J, Shao Y, Fu L, Goveas J, Ye E, Li W, Cohen AD, Chen G, Zhang Z. 2011.** Identification of hyperactive intrinsic amygdala network connectivity associated with impulsivity in abstinent heroin addicts. *Behavioural Brain Research* **216**:639–646 DOI 10.1016/j.bbr.2010.09.004.