# Peer review of "Morphological connectivity correlates with trait impulsivity in healthy adults"

_PeerJ, doi:10.7717/peerj.3533_

## Round 0.1 · original submission · Major Revisions

The reviewers raised questions regarding the parameters used for the analysis. The authors should report additional analyses to show the robustness of the reported correlations against variations in parameters/settings.

·

Basic reporting

The authors are suggested to further edit their language. There are numerous typos and grammar errors in the manuscript. Here I just list some of them.

1. Abstract section (Line 28). "was found correlated with" should be "was found to correlate with";

2. Abstract section (Line 37). "facilitating our understanding neural correlates" should be "facilitating our understanding of neural correlates";

3. Introduction section (Line 57). "associated gray matter" should be "associated with gray matter";

4. Introduction section (Lines 69-71). The sentence need to be rephrased.

5. Introduction section (Line 89). "The selection of ACC as seed points for three reasons" is not a sentence.

6. Materials and Methods section (Line 141). please see Kong (2014) (Kong et al., 2014).

7. Results section (Line 181). For each what?

8. Introduction section (Lines 73 and 75). Check the two references for their format. They are listed in the reference list as 2013a and 2013b.

Experimental design

1. The authors chose the AAL atlas to define ROIs. However, only association regions were used in their study. The authors should explain why they only concern the association regions. Further, why did the authors restrict their analyses to ipsilateral regions?

2. More details are needed for the MC calculation. Specifically, did the authors use the dartel method during T1 image segmentation? How much were the GM images smoothed? For the KDE estimation, how many sampling points were used? And whether the GM intensity boundary was fixed for all regions? In Figure 1B, it seems that the GM intensity boundary is set to [0 1].

Validity of the findings

No comment.

Additional comments

Kong and colleagues used a single-subject morphological connectivity (MC) approach to investigate the relationship between the ACC MC and impulsivity in 24 healthy adults. The manuscript is well and clearly written, the methods are sound, and the results are interesting. Specifically, their finding that the ACC-ITG MC was correlated with attentional impulsivity extends current literature regarding neural network for impulsive control, although the reproducibility should be examined in the future.

·

Basic reporting

No comment.

Experimental design

I couldn't find a clearly stated hypothesis in the introduction section of this paper. To me, it seems that the authors performed an exploratory study and reported what came out of it. I suggest that, at the end of the introduction section, the authors add a sentence stating a main hypothesis for their study or otherwise explicitly state the exploratory nature of their study.

Validity of the findings

Although, the study is done rigorously and the results are what they are and seem interesting, I still question the validity and the robustness of this kind of findings. As shown in my last paper (Van Schuerbeek et al. 2016), linking personality traits to brain structures using regression analyses seems to be very sensitive to "accidental" methodological choices made (increasing the risk for false positive results). Although, the authors did not use the VBM or ROI labeling method as I did, I do expect to see the same problems with robustness and repeatability of the results. It is not that I want to stop this kind of studies, since they could still lead to valuable insights, but given the general lage problem of false or at least unrepeatable results in this kind of neuroresearch, I would like to see that the authors add a separate section to the discussion in which they discuss the efforts they have done to test the robustness of their results for:
1. the sample used. -> Is there any other repositories available with which the study can be repeated or the sample size extended?
2. the regression model. -> what happens with the results if a parameter in the model is left out (e.g. age or sex) or added to the model?
3. ROI atlas used. -> the authors did the processing in SPM8 using the AAL atlas to define the ROIs. FreeSurfer was used to determine the total brain size. Do you find the same results if you use the ROI labeling results from FreeSurfer?
If the results are found to be robust for these methodological choices, it would certainly strengthen the credibility of the obtained results. Maybe, the authors did not test for these things, but they can at least discuss their expectations in the field.

As mentioned by the authors, their sample size was rather small. From the plot given in figure 2.B, it seems to me that the correlation found in the ACC-IFG MC is largely driven by one observation away from the others. I would like to suggest to the authors, to add a remark on this to their discussion about the problem of the sample size (line 278-281).

In general, when looking at the plots, the error on the correlation coefficients seems large. I suggest to the authors that they systematically report the standard error and the 95% confidence interval for all correlations in the results section.

Additional comments

No comments

---

## Round 0.2 · accepted · Accept

The revised manuscript adequately addresses reviewer comments and is suitable for publication in PeerJ.

·

Basic reporting

All my concerns have been addressed.

Experimental design

All my concerns have been addressed.

Validity of the findings

All my concerns have been addressed.

Additional comments

All my concerns have been addressed.

·

Basic reporting

no comment

Experimental design

no comment

Validity of the findings

no comment

Additional comments

The authors adequately cared about my previous comments about the robustness of their results for methodological choses. Their new analyses support and strengthen the reliability of the obtained results. As mentioned by the authors, the small sample size still weakens their results, but to me, it seems worth to publish this manuscript.